# The Role of Religion and Culture in Intergenerational Transnational Caregiving: Perspectives from Nigerian Christian Immigrants in Northern BC

**DOI:** 10.3390/bs15101383

**Published:** 2025-10-12

**Authors:** Chibuzo Stephanie Okigbo, Shannon Freeman, Dawn Hemingway, Jacqueline Holler, Glen Schmidt

**Affiliations:** 1Department of Health Sciences, University of Northern British Columbia, 3333 University Way, Prince George, BC V2N 4Z9, Canada; 2School of Nursing, University of Northern British Columbia, 3333 University Way, Prince George, BC V2N 4Z9, Canada; shannon.freeman@unbc.ca; 3Department of Social Work, University of Northern British Columbia, 3333 University Way, Prince George, BC V2N 4Z9, Canada; dawn.hemingway@unbc.ca; 4Department of History, Women’s Studies and Gender Studies, University of Northern British Columbia, 3333 University Way, Prince George, BC V2N 4Z9, Canada; jacqueline.holler@unbc.ca; 5School of Social Work, University of Northern British Columbia, 3333 University Way, Prince George, BC V2N 4Z9, Canada; schmidt@unbc.ca

**Keywords:** transnational caregiving, elder care, religion, culture, Nigerian immigrants

## Abstract

Background/Rationale: Transnational caregiving may be influenced by religious beliefs and cultural traditions that frame elder care as both a moral and religious obligation. While migration alters caregiving dynamics, religious teachings and cultural expectations remain central in guiding transnational caregiving practices. This study examines how Christian Nigerians who have immigrated to Canada navigate caregiving responsibilities within a transnational context, integrating their religion, cultural values, and the practical realities of crossing borders. Methods: This study employed a predominantly qualitative narrative approach, drawing on in-depth interviews with Nigerian Christian immigrants (*N* = 10) providing transnational care. Data collection involved a pre-interview survey and semi-structured interviews, providing the opportunity for participants to share their lived experiences. Thematic analysis was used to identify recurring themes related to the role of religion and culture in caregiving, ensuring a comprehensive exploration of participants’ perspectives. Findings: Caregiving is shaped by religious duty and cultural obligation, reinforced by biblical teachings and cultural values. Participants view elder care as a moral responsibility, tied to spiritual rewards and familial duty. Despite migration demands, family-based care remains preferred over institutional care, with social stigma attached to neglecting elders. Conclusions: Religion and culture remain integral to transnational caregiving practices, sustaining caregiving responsibilities despite migration-related realities. While religious teachings provide moral guidance and emotional support, cultural expectations reinforce caregiving as a collective and intergenerational duty. Policies and resources are needed that support transnational caregivers, ensuring they can fulfill their caregiving roles while adapting to new sociocultural environments. Policymakers should prioritize the implementation of policies and programs to support transnational caregivers, including family reunification measures, caregiving-related travel provisions, culturally tailored eldercare services, diaspora–local collaborations, organized caregiver support groups, and financial mechanisms such as tax incentives for remittances dedicated to elder care.

## 1. Introduction

Transnational caregiving refers to the provision of care across national borders, typically by migrants who maintain connections with family members in their country of origin ([7]; [17]). This phenomenon has become increasingly crucial in the context of globalization, migration, and aging populations ([10]; [50]; [91]). Transnational caregiving involves various forms of support, including emotional, financial, and practical assistance, often mediated through technology and long-distance communication ([8]; [49]; [9]). Caregiving practices are shaped by cultural expectations, religious beliefs, familial obligations, and the availability of support systems in both host and home countries ([87]). The well-being of transnational caregivers has been explored, bringing forward the tension they may experience as they navigate cultural and religious norms alongside the practical realities of their host societies ([8]; [68]). While this tension underscores the burden of caregiving, it also reveals gaps in understanding how caregivers reconcile these demands with their personal and familial obligations, particularly in the context of religious and cultural values that influence intergenerational caregiving practices.

### 1.1. Background

#### 1.1.1. Cultural and Religious Influences on Elder Care Practices

Culture encompasses shared values and traditions that influence caregiving expectations ([29]), while religion provides a moral and spiritual framework for understanding these experiences ([54]). Religion is a system of beliefs, practices, and moral values centered on the worship of a higher power or divine being, and it provides a framework for understanding caregiving experiences ([54]). Christianity, in particular, shapes caregiving through biblical teachings such as the command to “Honor your father and mother,” framing care as both a spiritual duty and cultural expectation ([13]; [35]; [95]). This religious–cultural convergence is reinforced by wider traditions that position caregiving as a moral obligation and expression of intergenerational respect ([4]; [37]; [84]; [99]).

Across traditions, filial piety and religious teachings emphasize elder care as a moral duty. Confucian filial piety presents it as an ethical and hierarchical responsibility ([20]; [38]; [62]), while Christianity highlights service, compassion, and devotion to God ([35]; [102]). Despite modernization, filial piety continues to shape elder care practices ([21]; [62]; [72]; [93]). Islam similarly situates caregiving as both a family responsibility and a religious act guided by Quranic and Hadith principles ([1]; [3]; [45]). These perspectives converge in treating elder care as a divine–ethical duty, though the religious rationale differs across traditions ([24]; [43]; [82]).

Nigerian Christians ground caregiving in both biblical principles and communal practice. Religious networks provide support, while prayer and scripture reinforce caregiving decisions ([31]; [76]). Providing financial or instrumental care is a visible demonstration of respect, strengthening family and social cohesion ([34]; [33]; [53]; [78]). However, modernization, migration, and shifting family structures complicate these expectations and may create tensions between cultural obligations and contemporary realities of caregiving activities ([72]; [91]).

Religious beliefs serve as obligations and as sources of strength, with caregivers drawing resilience from values of compassion and duty ([57]). Yet, when economic or practical constraints limit the ability to provide care, caregivers may face moral scrutiny and guilt ([60]; [89]; [93]). These dynamics illustrate how caregiving reflects broader values of respect, devotion, and continuity, while also adapting to evolving social landscapes ([64]).

#### 1.1.2. Religious and Cultural Motivations for Caregiving

While existing studies on religious motivation and caregiving do not specifically focus on Nigerian Christian caregivers, the framework of intrinsic and extrinsic religiosity offers a lens to understand how religiosity influences caregiver well-being and stress. Four types of religious motivation intrinsic (internalized religious commitment), self-determined extrinsic (personal values with external influence), non-self-determined extrinsic (external pressures), and amotivation (lack of intent), shape caregiving experiences ([75]). Intrinsic and self-determined extrinsic motivations support sustained caregiving, while non-self-determined extrinsic motivation may increase stress, and amotivation can lead to disengagement ([85]).

Intrinsic religiosity is linked to higher self-esteem and lower depression ([59]; [71]; [103]), whereas extrinsic religiosity, driven by external rewards or pressures, often correlates with anxiety and depression ([58]; [66]). These associations vary, with some studies reporting weak correlations ([58]). For Christian caregivers, balancing religious motivation with external demands can shape their caregiving resilience, burden, and overall well-being ([39]). Religiosity remains a strong predictor of psychological adjustment ([86]), and individuals high in both intrinsic and extrinsic religiosity tend to adjust better ([69]). Understanding these distinctions clarifies how religious beliefs influence caregiver resilience, burden, and well-being. Informal caregiving motivations stem from cultural and societal influences, along with personal and relational factors. Key motivators include reciprocity, altruism, moral responsibility, and the desire for peaceful longevity ([37]).

Cultural values, beliefs, and societal norms significantly shape caregiving motivations, with religion and culture serving as underlying factors ([100]). The concept of cultural self-identity emerges as an overarching explanatory factor, influencing cultural duty, obligations, and values ([100]). Personal and relational motivations encompass reciprocity, affection, family values, and caregiving obligations ([100]). Reciprocity shapes caregiving in Nigeria, where support is often mutual ([37]). This framing positions caregiving as a cultural responsibility to older adults, who are honored as protectors of traditions, ancestral links, and spiritual mediators ([32]).

#### 1.1.3. Religious and Cultural Connections in a Transnational Setting

Religion and culture shape migrant caregiving by providing coping mechanisms and preserving identity ([63]; [67]). Migrant religious organizations often arise to fill gaps in inclusivity within mainstream cultural practices through language, music, and worship ([6]; [36]). These spaces transmit values across generations and foster belonging, though secular host societies, they may emphasize cultural preservation over cross-cultural engagement ([19]). While such practices strengthen resilience, they can also create tensions when navigating differing norms ([74]; [80]).

The intergenerational transmission of religiosity reinforces elder care values through worship, moral instruction, and caregiving routines ([12]; [90]). Among migrant families, this process is often intensified, as religious participation supports coping with trauma and cultural adjustment ([51]; [61]). Religious spaces address concerns that mainstream groups may overlook, underscoring religion’s role in community building and emotional well-being ([55]; [63]; [92]). Religion can sustain transnational ties, providing emotional and social support during the process of migration and integration ([61]; [63]; [67]). Religious socialization, for example, has been found to play a more vital role than leisure activities, such as sports, in helping migrants process trauma and adjust to new environments ([51]).

Transnational caregiving involves balancing responsibilities across borders. Religious connection helps by offering emotional and practical support, reinforcing caregivers’ sense of purpose and connection ([81]). Religious identities and cultural connections facilitate the exchange of resources and values between home and host countries ([30]; [61]). Although these ties foster resilience, they can also heighten pressures by reinforcing obligations that are difficult to fulfill across distance ([91]). Caregivers often rely on prayer, sacred ceremonies, and visits to religious places to cope with stress and maintain resilience ([2]; [65]).

Religious community support is important for caregivers’ well-being, with friendships contributing to quality of life ([18]). Yet, many caregivers report lower satisfaction with the support from religious communities compared to non-caregivers ([65]). More context-specific approaches are needed to reflect diverse caregiving experiences. Integrating religious coping into mental health practice may enhance care and strengthen collaboration between professionals and religious communities ([2]; [65]; [81]). Further research should examine how these practices evolve across contexts and adapt to shifting cultural and social landscapes.

#### 1.1.4. Theoretical Framework: Cultural Relativism

Cultural relativism, rooted in anthropology, offers a lens for understanding caregiving practices as shaped by cultural, social, and moral norms (Figure 1). It emphasizes interpreting behaviors within their cultural contexts rather than judging them by external standards ([15]; [44]). This framework is particularly relevant in transnational caregiving, where norms and expectations vary across societies. In collectivist cultures, such as Nigerian Christian communities, caregiving is often seen as a communal responsibility grounded in interdependence and religious duty ([31]; [47]). Traditions like Ubuntu frame caregiving as a moral obligation tied to community well-being ([52]; [84]). In contrast, individualist societies prioritize autonomy and formal care systems ([41]; [79]).

Religious traditions further shape caregiving norms: Confucian filial piety emphasizes ethical duty and harmony ([20]; [72]), Judeo-Christian beliefs link caregiving to divine obligation and spiritual reward ([14]; [95]), Islamic teachings stress compassion and respect ([1]; [45]), and Hinduism frames it as a karmic duty rooted in reincarnation and family roles ([26]; [48]). In transnational contexts, caregivers often navigate conflicting cultural norms between home and host countries, highlighting caregiving as a dynamic, culturally contingent practice ([4]; [11]; [87]). These tensions underscore the importance of understanding caregiving as shaped by migration, globalization, and shifting family structures.

Cultural relativism also raises critical debates. While it promotes tolerance and challenges ethnocentrism, critics argue it may obscure power dynamics and gender inequalities or hinder universal caregiving standards ([56]; [88]; [94]). For example, caregiving norms in many societies disproportionately burden women, reinforcing gendered expectations that limit autonomy and well-being ([5]). Additionally, resistance to universal standards can complicate efforts to address global caregiving challenges, such as those arising from aging populations and migration ([28]).

Despite these limitations, cultural relativism remains a valuable framework for understanding caregiving as a culturally embedded practice. It highlights how caregiving is shaped by social norms, religious beliefs, and historical legacies, offering a more respectful lens on diverse caregiving experiences. At the same time, critiques of cultural relativism remind us to stay alert to ethical tensions, global caregiving disparities, and systemic challenges that may be obscured by cultural specificity ([40]; [96]; [101]).

Although prior studies have examined cultural expectations or religious motivations in caregiving, there is limited empirical research on how these two factors intersect in the lived experiences of Nigerian Christian immigrants. Much of the existing literature emphasizes general cultural obligations ([10]; [91]) or religious duty in isolation ([31]; [35]), but the mechanisms through which faith, scripture, and communal norms sustain long-distance elder care and how these practices are adapted in relation to Canadian elder-care systems remain underexplored. Studies of diaspora religion have highlighted identity and gender dynamics ([67]), yet they do not account for how intergenerational caregiving values are negotiated within migration. By focusing on the narratives of Nigerian Christians in a northern community in a Western Canadian province, this study addresses this gap and contributes new evidence on how caregiving values are sustained, transmitted across generations, and adapted despite the challenges of migration.

#### 1.1.5. Research Question

Guided by the framework of cultural relativism, this study focuses on how Nigerian Christian immigrants interpret and practice elder care across borders through the lens of their cultural and religious values. It explores how caregiving responsibilities are shaped by Christian teachings, moral obligations, and Nigerian cultural expectations. The central research question asks: “How do Nigerian Christian immigrants in northern BC integrate religion and culture into their transnational caregiving practices?” The study further describes how these caregivers adapt to the complexities of caregiving in a Canadian context, how intergenerational caregiving values are sustained, and how participants negotiate cultural continuity and change in a transnational setting.

## 2. Materials and Methods

### Data Collection and Analysis

A narrative approach ([22]; [23]; [83]) using thematic analysis ([16]) was undertaken to examine the influence of religion and culture on transnational caregiving among Nigerian immigrants to Canada. The study focused specifically on first-generation Nigerian Christian immigrants residing in northern communities in a Western Canadian province.

Participants were recruited using a combination of purposive and snowball sampling. Criterion sampling ensured participants met eligibility requirements: at least 19 years of age, first-generation Nigerian immigrants residing in a northern community in a Western Canadian province in Canada for at least one year, and with experience providing care to an elderly relative aged 65 years or older in Nigeria. Recruitment involved posters and short recruitment letters outlining the study’s objectives and voluntary nature. These were disseminated through Nigerian associations, religious groups, Nigerian social media groups, and other relevant networks in Northern BC, as well as through email outreach and word-of-mouth referrals. Religious leaders, community leaders, and association administrators helped circulate the invitation, while early participants referred peers and relatives who also met the study criteria ([77]). Embedding recruitment in trusted community spaces enhanced credibility and reduced barriers to participation. The recruitment process continued until thematic saturation was achieved, meaning that no new insights were emerging from additional interviews ([42]; [77]). Efforts to recruit individuals from non-Christian religious backgrounds were not successful, reinforcing the study’s focus on Nigerian Christians.

A pre-interview survey was conducted to gather foundational demographic, socioeconomic, and caregiving experiences as they relate to religion and culture from the study participants. This ensured a detailed understanding of the participants’ backgrounds. Narrative interviews, averaging one hour, provided in-depth insights into participants’ caregiving roles, motivations, and challenges. Through open-ended questions, participants were prompted to share their thoughts on how religion and culture impact their caregiving roles, discussing societal norms and family dynamics that influence caregiving responsibilities. Participants reflected on the cultural and religious norms that shaped their caregiving practices, highlighting how these influences impacted their sense of responsibility, their emotional experiences, and their interactions with family members. Participants were encouraged to describe their caregiving approaches before and after immigrating to Canada, sharing how their caregiving choices and responsibilities evolved as they adjusted to life in a northern Canadian context. This conversational and adaptable approach facilitated open discussions that illuminated how religion and culture interact to influence caregiving practices in transnational contexts, particularly in navigating longstanding approaches, expectations, and adapting to new caregiving environments. The interview guide that structured these conversations is provided in Appendix A.

Thematic analysis was used to explore participants’ narratives and identify recurring themes related to the role of religion and culture in caregiving ([16]). This method was selected for its adaptability and thoroughness in capturing both overt and subtle meanings within the data, making it particularly effective for examining the intricate intersections of religion, cultural expectations, and caregiving dynamics. The analysis adhered to [16]’s ([16]) six-phase framework, ensuring a methodical and detailed examination. The process began with an immersion phase, involving multiple readings of transcripts to become well-acquainted with the depth and nuances of each participant’s narrative. Initial codes were generated in an inductive manner, concentrating on both explicit content and underlying meanings linked to religious obligations, cultural expectations, and caregiving motivations. Through this iterative process, recurring elements were identified and categorized into distinct themes and subthemes.

To bolster reliability and depth, the coding process was conducted iteratively with several rounds of refinement and constant comparison to ensure themes reflected the diversity of participants’ experiences. Thematic analysis enabled identification of shared patterns while also recognizing context-specific viewpoints ([16]; [25]). Credibility was strengthened through cross-checking themes against transcripts and maintaining reflexivity throughout the analytic process to minimize bias ([73]). Coding was primarily inductive, but interpretation of the themes was informed by the study’s cultural relativist framework, which emphasized understanding caregiving within its cultural and religious context. This approach revealed both the endurance and adaptation of cultural norms in the transnational context, providing insight into the interaction of religion, migration, culture, and caregiving. Rigor was established through strategies to ensure trustworthiness, including prolonged engagement with the data, iterative coding, and refinement of themes ([46]; [70]).

The first author, a Nigerian-born woman of color, immigrant, mother, and registered social worker, conducted all interviews and primary analysis. Her personal journey of migration to Canada in 2014 and her professional experience in long-term care and immigrant community support shaped her sensitivity to participants’ narratives and awareness of the cultural and religious frameworks surrounding caregiving. These experiences enriched her ability to build rapport and interpret participants’ perspectives, but also introduced potential biases. To address this, reflexive journaling and continuous attention to participants’ voices were employed to ensure that interpretations remained grounded in the data rather than preconceived assumptions ([27]).

The reception of the study was generally supportive. Participants described the focus on religion and caregiving as timely and relevant, noting that they had not previously been asked to reflect formally on these issues. While some prospective participants expressed initial concerns about confidentiality and stigma related to caregiving choices, these were addressed through clear assurances of anonymity, pseudonyms, and the voluntary nature of participation. Overall, participants reported that sharing their narratives was both personally meaningful and beneficial for the wider Nigerian community in Canada. The study was conducted in accordance with the Declaration of Helsinki and approved by the University of Northern British Columbia Research Ethics Board (protocol 6009597; 23 August 2024).

## 3. Results

This research involved ten Nigerian immigrant caregivers living in a northern community in a Western Canadian province. Participants were selected using a mix of snowball and purposeful sampling methods to capture a range of diverse and insightful viewpoints on the dynamics of transnational caregiving related to religion and culture. Table 1 represents the participants’ demographics. The sample reflects an equal gender distribution of five women and five men. The age of the participants ranged from mid-20s to early 50s, reflecting a broad spectrum of caregiving experiences shaped by different stages of life. The educational attainment of participants was notably high, with seven participants having earned postgraduate degrees (master’s or PhD), while three have bachelor’s degrees. Moreover, participants represented a range of professional fields, including nursing, physiotherapy, social work, community services, and business analysis. These professional experiences offered a variety of perspectives for understanding and navigating duties, especially striking a balance between personal and professional caregiving responsibilities. Nine of the participants were married, and one participant was single. The study involved participants with varying lengths of stay in Canada (2 to 15 years). This reflects differences in integration into Canadian society and exposure to local caregiving norms and perspectives, allowing us to explore how caregiving roles evolve with time spent in the host country.

### 3.1. Overview of Themes and Subthemes

This section introduces the main themes and subthemes that capture how participants make meaning of transnational caregiving through religious and cultural lenses. Based on the thematic structure presented in Table 2, two overarching themes were identified: the role of religion in caregiving and the role of culture in caregiving. The religious theme includes three subthemes: caregiving as a moral and religious duty, spiritual and parental blessings as rewards for caregiving, and religious fulfillment in transnational caregiving. The cultural theme comprises four subthemes: cultural obligation and social expectations, intergenerational transmission of caregiving responsibilities, resistance to institutionalized care and cultural adaptation, and the role of family networks. The themes and subthemes that follow provide insight into the intersecting moral, social, and emotional dimensions of caregiving across borders, drawing directly on participants’ lived experiences.

### 3.2. The Role of Religion in Caregiving

Participants emphasized the influence of religious teachings in shaping their approach to elder care, viewing it as a religious duty closely tied to honoring their parents. For many, caregiving is deeply rooted in their religious beliefs and perceived as both a moral and spiritual obligation. This commitment remains strong even across borders, as participants continue to provide care from a distance, drawing strength from their religious beliefs to navigate the emotional and logistical challenges of transnational caregiving.

### 3.3. Religious Teachings Framed Caregiving as a Moral Obligation

Participants framed caregiving as a religious duty, guided by biblical teachings. Religious values reinforced their commitment to elder care, even across borders, as they viewed caregiving as an act of obedience to God. P10, Male, emphasized the biblical foundation of caregiving: “I’m a Christian by practice, and the Bible speaks about honoring our parents… taking care of our parents, and I’m a strong believer of that.” Similarly, P7, Male, described caregiving as an extension of his religious beliefs: “I naturally also feel obligated to do that for them… The next big thing that has influenced my decision to take care of my folks is religion.” For some, religious teachings shaped their early understanding of caregiving as a moral responsibility, which they carried with them even after migrating. P1, Female, linked her motivation to her Catholic upbringing: “Growing up as a Catholic, I was taught that it’s your responsibility to help others… It doesn’t matter if I know you, if you’re in need, it becomes my responsibility.” P5, Female, also reflected on how religious beliefs influenced her caregiving role: “I do believe that those religious teachings have actually influenced how I have come to believe about taking care of my parents.”

Religious beliefs reinforced a sense of duty in long-distance caregiving. P3, Female, highlighted biblical teachings on obedience: “There’s definitely a lot of ‘obey your parents,’ and this is what Christ would like…” For P8, Male, religion also reinforced cultural caregiving values, strengthening their sense of responsibility even while living abroad. P8, Male, noted: “Religion has just come to reposition the value of our ancestors… to take care of their loved ones as the Bible instructs.” This suggests that for many caregivers, religious teachings did not replace cultural expectations but rather reinforced them, creating a dual sense of duty that persisted across borders. By framing caregiving as a religious obligation, participants found ways to remain actively involved in their parents’ care despite being geographically distant. Religious beliefs not only shaped their motivation to provide care but also influenced how they provided support transnationally.

### 3.4. Spiritual and Parental Blessings as Rewards for Caregiving

Caregiving was not just a duty but also a source of blessings, reinforcing commitment to elder care despite the demands of transnational caregiving. P8, Male, emphasized the rewards: “There is a blessing that comes with it.” Parental blessings were particularly significant. P4, Female, shared: “My mother prayed for me, she said, ‘What you did for me, your child will do for you.’” P2, Female, viewed prayers as encouragement: “I always saw it as a kind of blessing… Just their prayers and the fact that they would say ‘thank you’ would somehow help you out in the future” [implying that parental prayers were believed to bring divine favor and future well-being].

Biblical teachings reinforced the belief that honoring parents brings divine favor. P3, Female, stated: “One of my religious beliefs is that if you honor your father and mother, you will live a long life.” P8, Male, added: “The Bible says honor your parents so that it may be well with you.” Certain participants found lifelong reassurance in these beliefs. P6, Male, reflected on his father’s final words: “Before my dad passed on, he said, ‘My God will bless you.’ Those words are eternal.” He further emphasized the biblical promise: “The Bible says if you honor your parents, your days will be long.” These narratives illustrate how religious teachings shaped caregiving for these participants as both a moral duty and a pathway to spiritual and familial rewards. The principle of honoring one’s parents, deeply embedded in biblical and cultural traditions, was reflected in their lived experiences. It motivated care through the promise of divine favor and future blessings. For these participants, such beliefs offered ongoing reassurance and reinforced their commitment to elder care, even when faced with distance, pressure, and sacrifice.

### 3.5. Fulfillment in Transnational Caregiving

Participants described a deep sense of fulfillment rooted in their religious values, reinforcing their commitment to elder care despite geographical distance. P1, Female, linked this fulfillment to biblical teachings on kindness: “The Bible encourages us to care for others… It’s not all about you, it’s about kindness, and that gives me a sense of fulfillment.” This highlights how caregiving aligns with both personal and religious values, providing emotional and moral satisfaction beyond its practical responsibilities. P9, Male, expressed a similar sense of fulfillment: “The fulfillment … brings a lot of relief and a lot of joy …. Knowing also that not only am I able to support my parents but also support the people that are caring for them financially, emotionally, even with the level of the knowledge I have had since coming to Canada about caregiving.” P9, Male, found reassurance in knowing his parents were well cared for, regardless of location: “Those are other fulfillment[s] that I found in that. Knowing fully well that whether I’m in Nigeria or here in Canada, my parents are adequately cared for, and they are being taken care of.” These narratives demonstrate how caregiving, shaped by religious beliefs, provides a sense of purpose, emotional comfort, and moral reward, sustaining participants’ involvement in elder care across borders.

### 3.6. The Role of Culture in Caregiving

Participants recounted how culture played a vital role in their motivation to provide care and support to family members transnationally. The following themes—obligation and social expectations, intergenerational transmission of caregiving responsibilities, resistance to institutionalized care and cultural adaptation, and family networks—emerged from culture as it relates to transnational elder care for Nigerian elderly.

### 3.7. Cultural Obligation and Social Expectations

In Nigerian cultures, caregiving is a shared duty embedded in societal values and expected across religious and ethnic lines. P4, Female, emphasized: “Whether you’re Muslim, Christian, or a pagan worshipper, it just doesn’t matter. That aspect of life [caring for one’s parents]is something that cuts across all ethnic groups and religions in Nigeria.” This obligation extends beyond presence, shaping how care continues from abroad. P10, Male, stated: “Culture is a vital component of caregiving. In Africa, caregiving is embedded into our culture. You grow up knowing that it’s your responsibility to care for your parents.” P9, Male, added: “Our culture pays homage and holds in high regard children who take care of their parents.” Even at a distance, caregiving remains essential, with migrants ensuring parental well-being through financial and emotional support.

Failing to provide care carries consequences. P3, Female, noted: “If you don’t do what is expected, you’re being discriminated against.” P9, Male, reinforced this: “As a child, for instance, if you are not able to take care of your parent, society frowns at it. In fact, it may be considered taboo.” The expectation to provide care remains strong, even when migration limits physical involvement. Social exclusion can follow neglect. P7, Male, stated: “If you don’t look after your parents, from my culture, people may excommunicate you. You may be excommunicated from your community, and people may start keeping distance from you.” P9, Male, explained: “If parents are growing old and their children are not taking care of them, it will be considered an act of irresponsibility. That’s the way our culture views it.” These narratives show that migration does not erase caregiving duties but reshapes them, with transnational caregivers fulfilling obligations through ongoing support despite physical separation.

### 3.8. Intergenerational Transmission of Caregiving Responsibilities

Caregiving is passed down through generations, shaping obligations that persist across borders. Participants described learning caregiving practices by observing their parents, reinforcing caregiving as both a familial and cultural expectation. P10, Male, explained, “I’ll say it’s something that was instilled in me because I saw them do it for their own folks, so I naturally also feel obligated to do that for them.” P6, Male, shared, “I watched my dad take care of his mother… that encourages you to do the same.” P2, Female, reflected on caregiving as a generational responsibility: “When she did that, and she’s speaking to me from her lived experience of taking care of her mother-in-law and her dad, I just felt it’s something I should also take after as well.”

Caregiving is also modeled for younger generations, reinforcing long-term expectations despite distance. P6, Male, noted, “My kids are also looking at me to do it for my parents… they are really learning.” He added, “We do it so that as we are doing it, our own children would see that we are involved in taking care of our own parents. So that in essence, they will learn it.” P4, Female, connected this to cultural teachings: “There’s an adage in Igbo land that says that when a parent finishes training a child, the child trains the parents, and it relates to caregiving.” These narratives highlight how caregiving remains a transgenerational duty, shaping the way participants continue to care for aging parents from afar. Even in transnational contexts, caregiving is sustained through financial, emotional, and logistical support, ensuring that cultural expectations are upheld across generations.

### 3.9. Resistance to Institutionalized Care and Cultural Adaptation

Participants highlighted the contrast between caregiving practices in Nigeria and Canada, emphasizing the strong preference for family-based elder care in Nigerian culture. Placing aging parents in care homes is widely considered unacceptable. P3, Female, described her reaction to Canadian practices: “It was a culture shock for me seeing a lot of elderly people in homes, as back home we don’t have homes for them.” P4, Female, reinforced this view: “It would be unheard of to place your parents in a home. It’s considered a taboo in our culture.” Maintaining direct family involvement in elder care remains a priority, even in transnational settings. P5, Female, explained, “From my culture, it’s taboo for you not to take care of your elderly ones. In Canada here, once your parents are aging, you can take them to the Old People’s Home.” P2, Female, added, “So it’s a cultural thing, even if they are staying alone, there would always be like someone that would take care of the person, who is either a close relative or a paid caregiver. But care homes are not common. You either do it or you get someone who can help you take care of the elderly.”

While some participants acknowledged the social benefits of institutional care, such as opportunities for older adults to socialize, they still prioritized family oversight. P2, Female, shared, “I would like my father-in-law to go to a care home just to socialize with others, but I still think it’s important to have someone closely connected to the family to keep an eye on him.” P8, Male, emphasized cultural expectations against institutionalized care: “In our community, people are expected to care for their older adults within the home. Sending an elderly parent to a care facility is often frowned upon.” These perspectives illustrate how Nigerian cultural norms shape caregiving expectations, even across borders. Despite exposure to different elder care systems in Canada, participants continue to prioritize family involvement, adapting caregiving strategies to uphold culturally rooted values while managing transnational responsibilities. This ongoing preference is not simply due to a lack of long-term care infrastructure in Nigeria but rather reflects deeply ingrained cultural norms that value home-based elder care. Even after migration, Canadian models of institutional care have limited influence, as participants maintain strong commitments to family-centered caregiving shaped by cultural expectations.

### 3.10. Extended Family Involvement in Caregiving

Caregiving is characterized by collective effort within the extended family. P5, Female explained, “I had other family members step in to assist me too,” illustrating the shared nature of caregiving responsibilities. P1, Female further elaborated on the communal aspect of caregiving, noting, “In Nigeria, caregiving is more communal… It’s not just on one person; the family comes together to help.” These reflections demonstrate how family networks distribute caregiving duties, easing the burden on any single individual. While caregiving was a shared responsibility, P7, Male also noted that this structure remains influential even in a transnational context, where those living abroad rely on family networks back home to provide direct care. He explained, “We have to ensure other systems are put in place, so that it becomes a smooth aging process for my family, and at the same time, being able to enhance the life of those at home who now play the roles of caregiving.” These accounts highlight how caregiving is sustained through collective family efforts, ensuring that older adults receive support both from those present and from relatives living abroad who contribute in various ways.

## 4. Discussion

This study examines how religion and culture shape caregiving practices among Nigerian immigrants in a northern community in Western Canada. Using cultural relativism, caregiving is understood as embedded in cultural, social, and religious frameworks that intersect with migration to influence responsibilities and decisions. Among participants, caregiving is upheld by Christian ethics as both a sacred duty and a social responsibility, rooted in biblical commandments to honor father and mother. Christian beliefs provided a central motivation, framing caregiving as obedience to God and reinforcing it as a spiritual practice that extends across borders despite the challenges of transnational caregiving.

The findings align with research identifying religion as both emotional support and a framework for cultural expectations in caregiving ([48]; [57]; [98]). Religious teachings frame caregiving as a moral and divine obligation that emphasizes compassion, respect, and responsibility ([1]; [45]). These frameworks are culturally embedded, differing across contexts yet equally central within their communities ([56]; [94]). Prior studies also show that religious teachings serve not only as a familial duty but also as a way of preserving social cohesion and moral integrity ([35]; [37]; [84]).

Culture also plays a central role in shaping transnational caregiving among Nigerian immigrants, with practices sustaining cultural identity even when care is provided across borders. This dynamic mirrors other cultural contexts, such as East Asian societies, where filial piety positions elder care as both a moral duty and a source of pride ([20]; [72]). Similarly, transnational caregivers adapt traditional caregiving models to the realities of migration while maintaining cultural values ([50]; [91]; [97]).

Caregiving responsibilities are viewed as intergenerational expectations, passed down through role modeling. Parents in Canada demonstrate caregiving to their children, transmitting values across generations. Participants observed their parents caring for elders and saw this as a blueprint for their own caregiving roles. This reflects findings that observed family caregiving sustains practices across generations ([78]; [84]). Such continuity reinforces communal values of reciprocity and interdependence ([56]; [94]), contrasting with individualistic models where formal care systems often replace familial duties ([79]). Migration complicates these dynamics, requiring adaptation of longstanding practices to new socio-cultural settings ([4]; [17]; [76]).

A recurring theme was the contrast between family-based care in Nigeria and institutional care of older adults in Canada. Participants expressed a preference for family caregiving, consistent with Nigerian norms that view care homes as taboo, though some participants acknowledged the social benefits of communal housing. There was no clear link between the length of residence in Canada and greater acceptance of institutional care. While migration created tension between caregiving obligations and geographic distance, participants did not interpret absence as neglect. Instead, they adapted by offering financial, emotional, and logistical support from afar. This reflects the cultural significance of familial elder care in African societies ([53]). Even when recognizing the value of Canadian care homes, participants stressed the need for family involvement. Cultural relativism contextualizes these preferences within Nigerian caregiving values of reciprocity and communal responsibility. While migration may have newly introduced institution-based care models to participants, they described commitment to their own cultural norms. In practice, this led to some participants adopting a hybrid approach, blending Nigerian cultural expectations with selected Canadian practices while still prioritizing home-based care as a moral and cultural duty ([10]).

While caregiving was consistently framed as a moral and religious duty, participants described subtle tensions between cultural expectations and the realities of caregiving in Canada. These tensions were shaped not only by migration but also by differences in household structure, professional background, gender roles, and length of residence. For example, participants who had resided in Canada for more than 5 years expressed more nuanced views on institutional care, while those newer to the country emphasized traditional models. Professional caregivers often described a dual burden of formal and familial care, highlighting how occupational roles and gendered expectations intersect with cultural obligations. These contrasts suggest that caregiving adaptation is not uniform but shaped by layered factors that merit further comparative analysis.

Although this study provides valuable insights into the role of religion and culture in shaping transnational caregiving practices among Nigerian immigrants in Canada, several limitations. While these participants provided rich qualitative insights, the focus on caregivers in a limited geographical area in one Western Canadian province offers a context-specific lens into caregiving experiences. This was expected as it reflects the demographic reality of the community where Christianity is the predominant religious affiliation. The sample’s homogeneity- comprising exclusively Christian and highly educated participants narrows the scope of the study. The alignment with the exploratory and descriptive focus of this work, where broad generalizability was not the focus. Instead, the aim was to describe in-depth how caregiving is shaped within a particular social, cultural, and geographical setting. Further research is encouraged to expand upon this foundational work and further explore more diverse experiences of transnational Nigerian caregivers from different geographic, religious, socio-economic, and educational backgrounds.

There is a great opportunity to expand on findings from this study by future inclusion of the perspectives of older adults receiving care and support from their transnational caregiver(s) living abroad. This would shed light on whether older adults’ perspectives on the influence of migration on caregiving practices are similar or different from those of their transnational caregivers’ expectations and cultural values.

## 5. Conclusions

This study describes the roles of religion and culture in shaping transnational caregiving among Nigerian Christian immigrants in a northern community in a Western Canadian province. Caregiving remains deeply tied to cultural and religious values, sustaining obligations that continue across borders. While migration presents new caregiving dynamics, family networks, religious beliefs, and cultural traditions shape how care is provided, whether through financial support, emotional connection, or remote coordination. Findings suggest that caregiving is not only a personal duty but also a communal and intergenerational responsibility, reinforced through role modeling and religious teachings. Despite reservations about elder care facilities, participants recognized the benefits of integrating family involvement with formal support systems. These findings point to the need for policies and programs that support transnational caregivers, such as family reunification measures, caregiving-related travel provisions, and culturally tailored eldercare services.

### 5.1. Significance and Implications of the Study

The findings show that while Nigerian immigrants navigate geographical distance, they adjust transnational caregiving to align cultural expectations with migration realities, maintaining caregiving as a cultural and religious responsibility. Family members collectively support older adults, reducing the burden on individuals but requiring coordination from those abroad. As these caregiving practices continue across borders, there is a need for policies and resources that reflect these values. Religious organizations and cultural institutions could play a crucial role in reinforcing this communal model by facilitating resource-sharing and building transnational networks that support caregivers’ efforts. For example, organized support groups for Nigerian caregivers in Canada could serve as hubs for exchanging strategies, providing emotional support, and mobilizing financial assistance for elder care in Nigeria ([63]; [67]). Such initiatives could also include diaspora-led caregiving workshops or financial support programs to ease the burden of cross-border caregiving.

The communal approach is a defining feature of Nigerian caregiving practices, but migration can complicate this culturally rooted model. Many caregivers in Nigeria rely on extended family to share responsibilities, reinforcing the need for strong transnational ties. Yet, sustaining elder care across borders is fraught with tension, requiring constant negotiation and adaptability. Maintaining these connections ensures that caregiving duties are distributed among family members, even when some live abroad. However, distance can limit direct involvement, making it necessary to find alternative ways to stay engaged. Policies and programs that support family reunification, visits, or remote coordination could strengthen these efforts, allowing caregivers to fulfill their roles while navigating the complexities of transnational caregiving.

Migration exposes caregivers to additional resources for care for older adults, such as institutionalized care, which is not widely available in Nigeria. Institutional care contrasts with the culturally ingrained preference for family-based caregiving in Nigerian society. While participants recognized the potential social benefits of care homes, they highlighted the importance of maintaining family oversight. This openness suggests that hybrid caregiving models integrating elements of family and institutional care could be explored. It also reflects a shift in long-standing caregiving expectations, showing how culturally rooted perspectives may adapt to accommodate new demands, resources, and realities. Policymakers and service providers could facilitate these hybrid approaches by offering culturally sensitive eldercare services that combine communal caregiving values with formal support structures.

### 5.2. Recommendations and Future Directions

Future research should adopt more diverse and representative samples, including caregivers from different religious, socio-economic, and geographic backgrounds. Incorporating the perspectives of older adult care recipients, as well as the incorporation of sex- and gender-based analysis, would provide deeper insight into transnational caregiving. Longitudinal studies could examine how migration and cultural adaptation influence caregiving roles over time. Comparative research across cultural and religious groups would further illuminate commonalities and differences in caregiving practices, strengthening understanding of how cultural and religious frameworks intersect with transnational responsibilities. Participants did not consistently recognize that in Canada, most older adults live at home and prefer to age in place rather than in a long-term care facility. Comparative designs can disentangle cultural perspectives from system features and show when hybrid models arise.

In addition to research directions, this study identifies several policy and practice recommendations. Policymakers should develop programs that directly support transnational caregiving, including family reunification measures and caregiving-related travel provisions, as well as culturally tailored eldercare services. Nigerian diaspora organizations could collaborate with local institutions to establish eldercare programs in Nigeria that combine professional care with strong family connections. In Canada, culturally sensitive services for older adults and organized support groups for transnational caregivers could provide emotional support, resource-sharing, and financial guidance. Tax incentives for remittances directed toward elder care could further ease the financial strain of transnational caregiving. Collectively, these measures would help sustain caregiving as both a cultural and religious responsibility while reducing the challenges faced by caregivers navigating obligations across borders.

## Figures and Tables

**Figure 1 behavsci-15-01383-f001:**
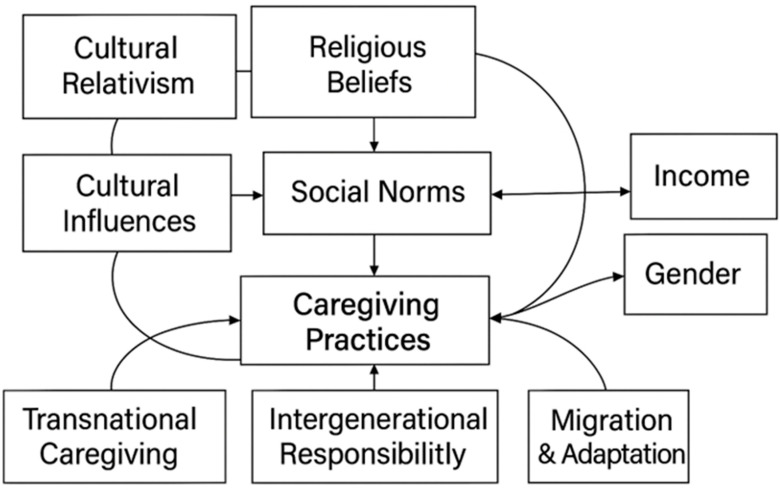
Cultural Relativism as a Framework for Understanding the Influence of Culture and Religion in Transnational Caregiving.

**Table 1 behavsci-15-01383-t001:** Demographics of Study Participants.

Participants	Age Range	Gender	Marital Status	Years in Canada	Region of Origin	Education Level	Religious Importance
**1**	35–44	Male	Married	9	South-East	Bachelor’s Degree	Important
**2**	35–44	Female	Married	8	North-Central	Master’s Degree	Very Important
**3**	35–44	Female	Married	3	South-East	PhD	Important
**4**	45–54	Male	Married	12	South-South	PhD	Very Important
**5**	35–44	Female	Married	5	South-East	Master’s Degree	Important
**6**	35–44	Male	Married	3	South-West	PhD	Very Important
**7**	45–54	Female	Married	2	South-East	PhD	Very Important
**8**	25–34	Female	Single	5	North-Central	Master’s Degree	Not Significant
**9**	45–54	Male	Married	15	South-West	Bachelor’s Degree	Very Important
**10**	35–44	Male	Married	3	South-South	Bachelor’s Degree	Very Important

**Table 2 behavsci-15-01383-t002:** Overview of the main themes and sub-themes.

*Theme*	*Subtheme*	*Key Findings*	*Quotes*
*The Role of Religion in Caregiving*	Caregiving as a Moral and Religious Duty	Religious teachings and biblical commandments reinforce caregiving as an act of obedience to God.	“I’m a Christian by practice, and the Bible speaks about honoring our parents… taking care of our parents, and I’m a strong believer of that.” (P10, Male)
	Spiritual and Parental Blessings as Rewards for Caregiving	Caregiving brings spiritual rewards, including divine favor and parental blessings.	“My mother prayed for me, she said, ‘What you did for me, your child will do for you.’” (P4, Female)
	Religious Fulfillment in Transnational Caregiving	Providing elder care brings emotional and moral fulfilment, aligning with religious values.	“The Bible encourages us to care for others… It’s not all about you, it’s about kindness, and that gives me a sense of fulfilment.” (P1, Female)
*The Role of Culture in Caregiving*	Cultural Obligation and Social Expectations	Caregiving is deeply embedded in Nigerian culture, seen as a shared family duty across religious and ethnic lines.	“Culture is a vital component of caregiving. In Africa, caregiving is embedded into our culture. You grow up knowing that it’s your responsibility to care for your parents.” (P10, Male)
	Intergenerational Transmission of Caregiving Responsibilities	Caregiving is passed down through generations, with individuals learning caregiving responsibilities from their parents.	“I watched my dad take care of his mother… that encourages you to do the same.” (P6, Male)
	Resistance to Institutionalized Care and Cultural Adaptation	Participants strongly preferred family-based elder care, viewing care homes as culturally unacceptable, though some acknowledged benefits.	“It was a culture shock for me seeing a lot of elderly people in homes, as back home we don’t have homes for them.” (P3, Female)
	Family Networks	Caregiving is a collective responsibility. Extended family members share caregiving duties, ensuring support across transnational settings.	“In Nigeria, caregiving is more communal… It’s not just on one person, the family comes together to help.” (P1, Female)

## Data Availability

The dataset presented in this study is securely stored on OneDrive, managed by the UNBC secure server. Access to the data may be granted upon reasonable request and with appropriate ethical approval. Requests for access should be directed to the corresponding author, Chibuzo Stephanie Okigbo.

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
