# Peer review of "The Role of Religion and Culture in Intergenerational Transnational Caregiving: Perspectives from Nigerian Christian Immigrants in Northern BC"

_behavsci, 2025, doi:10.3390/bs15101383_

Round 1
Reviewer 1 Report
Comments and Suggestions for Authors
The article offers a clear and well-articulated contribution, based on a solid theoretical and empirical literature base. Citations include both foundational and recent works, showing strong grounding in current academic discourse, and the range of references is broad and interdisciplinary. The authors have done a good job integrating definitions of the core variables under study, and the inclusion of comparisons with other religious traditions is both relevant and well-executed.
The exploration of the interplay between religion, culture, and migration is timely and engaging. The research question is clearly stated, and the study design is appropriate. The narrative approach is well-suited to the topic, and the methodology section is generally clear. The results section is well-organized, with tables that help summarize key points, and the discussion follows the logic of the theoretical background, which makes the paper coherent. The conclusion includes implications that speak to both academic inquiry and policy development.
That said, there are several areas where the paper could be improved. I list them below:
- Most claims are generally well-supported, though a few paragraphs in the Theoretical Background could benefit from less reliance on one or two sources (e.g., Esiaka & Luth) to avoid overdependence.
- The literature review and discussion sections could benefit from tighter editing to avoid repeating the same points with different citations.
- I would suggest moving the theoretical framework about cultural relativism from the materials and methods section. It could be embedded into the previous background sections, or a new subsection in the background should be added to present that. Either way, it has to come sooner. The section also could be shortened, as it recalls some theory origins (ex.: discussion of Franz Boas and the historical origins of cultural relativism) which are not necessary. This section is a bit repetitive itself, but also in comparison with the other subsections from the background.
- The literature gap that is being filled could also be better ellucidated. The authors do a good job in articulating the cultural context and the other investigated variables, but it would benefit from having a clearer gap that is supposedly being addresed.
- The interview guide or sample questions should be included, either as an appendix or described in the methodology.
- There must be more information on how the participants were recruited, and if possible, how the theme was received by them. For example, how were participants contacted? How was the study received in the community?
- The last paragraph of the data collection section could be shorter. It gets a bit repetitive when presenting the thematic analysis assumptions.
- There is a sentence that says "Nine of the participants were married, and one participant was single, which adds depth to the analysis". How exactly does that add depth, as there is almost no variance of civil status?
- The results section would benefit from a more explicit rationale on what theories and assumptions guided the coding. Was the thematic analysis entirely inductive? Were any theories used to guide code development?
- The discussion occasionally reiterates points made earlier, and could benefit from a more critical analysis of contradictions or tensions (e.g., where cultural values may limit adaptation to host country care systems). For example, why did you not consider gender as a variable of analysis, specially because the literature shows that caregiving is so gendered? One potential enhancement would be greater cross-comparison among participant responses (ex.: comparing perspectives by gender, years in Canada, or religious intensity).
- The qualitative analysis could benefit from a positionality statement. How did authors' own backgrounds influenced both the interviews and the interpretations?
Author Response
We are grateful for your thorough review and valuable comments on our manuscript. Your observations and suggestions have been instrumental in improving the quality and clarity of our work. Below, we provide detailed responses to each point raised and outline the changes that have been made to the manuscript. All changes are highlighted in red in the revised document. We hope that the revisions meet your expectations and improve the rigour and clarity of the paper.
|
ID |
Comment |
Response |
|
Comments From Reviewer 1 |
||
|
R1-01 |
Most claims are generally well-supported, though a few paragraphs in the Theoretical Background could benefit from less reliance on one or two sources (e.g., Esiaka & Luth) to avoid overdependence. |
We appreciate this insight and have added additional references to the background section as suggested. |
|
R1-02 |
The literature review and discussion sections could benefit from tighter editing to avoid repeating the same points with different citations. |
We have carefully edited these sections as suggested by the reviewer. |
|
R1-03 |
I would suggest moving the theoretical framework about cultural relativism from the materials and methods section. It could be embedded into the previous background sections, or a new subsection in the background should be added to present that. Either way, it has to come sooner. The section also could be shortened, as it recalls some theory origins (ex.: discussion of Franz Boas and the historical origins of cultural relativism) which are not necessary. This section is a bit repetitive itself, but also in comparison with the other subsections from the background. |
We appreciate this suggestion and have moved cultural relativism to Background, trimmed historical detail, and sharpened the link to caregiving. |
|
R1-04 |
The literature gap that is being filled could also be better ellucidated. The authors do a good job in articulating the cultural context and the other investigated variables, but it would benefit from having a clearer gap that is supposedly being addressed. |
We agree with this comment and subsequently, we added additional detail to enhance clarity in the gap. Please see lines 204-216. |
|
R1-05 |
The interview guide or sample questions should be included, either as an appendix or described in the methodology. |
The Methods section now notes the interview guide has been included as an Appendix. |
|
R1-06 |
There must be more information on how the participants were recruited, and if possible, how the theme was received by them. For example, how were participants contacted? How was the study received in the community? |
We have provided additional detail as requested. See lines 228-250 and 302-309 |
|
R1-07 |
The last paragraph of the data collection section could be shorter. It gets a bit repetitive when presenting the thematic analysis assumptions. |
We revised the final paragraph to improve clarity and remove repetition. |
|
R1-08 |
There is a sentence that says "Nine of the participants were married, and one participant was single, which adds depth to the analysis". How exactly does that add depth, as there is almost no variance of civil status? |
We appreciate this keen insight to detail and have removed this comment. Marital status is now only noted Table 1. |
|
R1-09 |
The discussion occasionally reiterates points made earlier and could benefit from a more critical analysis of contradictions or tensions (e.g., where cultural values may limit adaptation to host country care systems). For example, why did you not consider gender as a variable of analysis, specially because the literature shows that caregiving is so gendered? One potential enhancement would be greater cross-comparison among participant responses (ex.: comparing perspectives by gender, years in Canada, or religious intensity). |
There has been a lot of revision to the methods to make it clearer for the reader that the inductive coding and interpretation was guided by cultural relativism. As sample size was limited it is difficult to make comparisons by gender, years in Canada etc. We have noted this in the limitations section and encourage future research to examine this further. See lines 574-583. |
|
R1-10 |
Discussion repeats earlier points and should analyze contradictions (e.g., culture vs. Canadian systems, gender). |
The discussion section was revised to highlight tensions, cross-compare perspectives (gender, household structure, years in Canada, professional background) and remove duplication from other parts of the study. |
|
R1-11 |
The qualitative analysis could benefit from a positionality statement. How did authors' own backgrounds influenced both the interviews and the interpretations? |
This has been added to the method section as requested. See lines 294-302. |

Reviewer 2 Report
Comments and Suggestions for Authors
Thank you for the opportunity to review such an important piece of work. I enjoyed reviewing your manuscript. Detailed remarks, suggestions, and comments are contained in the attached reviewer report.

Author Response
We are grateful for your thorough review and valuable comments on our manuscript. Your observations and suggestions have been instrumental in improving the quality and clarity of our work. Below, we provide detailed responses to each point raised and outline the changes that have been made to the manuscript. All changes are highlighted in red in the revised document. We hope that the revisions meet your expectations and improve the rigour and clarity of the paper.
|
ID |
Comment |
Response |
|
Comments From Reviewer 2 |
||
|
R2-01 |
Your abstract is well-written and provides a concise overview of the manuscript's contents. To add more value, consider briefly indicating a recommendation that emerged from this study. Your keywords are appropriate and relevant to the manuscript. |
Thank you for this suggestion. We have added this to the abstract as suggested. See lines 33-37. |
|
R2-02 |
It would be beneficial to include a reference to the first statement in the introduction (Page 1) |
Added citations: Baldassar et al. (2006); Baldassar & Merla (2014). |
|
R2-03 |
Also, add a reference to your definition of “the transmission of religiosity across generations” – It is the first statement on page 5. |
Added citations: Bengtson et al. (2009); Silverstein et al. (2023). See line 135-137 |
|
R2-04 |
You can add more value by also indicating the research questions at the end of your introduction, which will then nicely lead to the materials and methods section (NB this is currently in the methodology section). |
The research focus heading was changed to Research Question and has been placed where the reviewer has suggested. See line 217-227. |
|
R2-05 |
Whilst your methodology is solid and well written, I feel that pages 6 to 8 (theoretical framework: Cultural relativism; research focus) belong to the introduction section. You can keep them as they are and just shift the heading 2: Materials and Methods to page 8, just after the research focus. See whether you agree with this suggestion. I think it improves the flow of your work. If you don't agree with it, it is also okay. |
As noted above in our response to reviewer 1 (R1-03), we have moved cultural relativism to Background, trimmed historical detail, and sharpened the link to caregiving. |
|
R2-06 |
Discuss the population, sampling, type of research, research approach, and ethical considerations (NB some of this information is currently presented in the results section). Although towards the end of your manuscript, you state that this study received ethical clearance, this statement should also appear in the methodology section; you can literally copy and paste it. |
As noted above in our response to reviewer 1 (R1-06), we have provided additional detail as requested. See lines 228-250 and 302-309. Ethics statement moved see 309-312. |
|
R2-07 |
The first statement in this section should be shifted to the methodology section. See whether you agree (see my comments above on what I felt was lacking in the methodology section). Should you not agree to this suggestion, it is still okay since it is not a big deal. |
We appreciate this suggestion and have moved this to the methods section as suggested. |
|
R2-08 |
Your discussion section is well presented. I am impressed with how you managed to integrate information from the study with your literature review. This section clearly shows how your findings fit in the broader literature on the topic. Well done, keep up the good work. The implications, limitations, and future direction sections are well explained. I usually prefer these sections to come under the conclusions section as subheadings. I find that it brings a clearer and logical flow. This is just a suggestion; you don’t have to follow it. Just see where these sections are better placed under the discussion or conclusion sections. Other than that, these sections are solid, well-done. |
Thank you for your kind words. We have reflected on the organization of the paper and section and refocused in accordance with the suggestions from R1 and R2. We hope these changes will meet R2’s approval. |
|
R2-09 |
Conclusions Your conclusion section is solid. The recommendations are already in this section; see whether it is possible to present them as a subheading so that they are better illuminated. The following is just a suggestion; see whether you want to adopt it or not. 5. Conclusion – here you provide your traditional conclusion, like what you have done 5.1. Significance and implications of the study – NB This section is currently under the discussion section. 5.2. Limitations – NB This section is currently under the discussion section. 5.3. Recommendations – here you can indicate the recommendations that emerged from your study. NB This section is currently integrated into the conclusion section. If the word count allows you, you can revisit your limitations section to deduce more recommendations. |
We have restructured the conclusion section but kept the limitations at the end of the discussion as the flow fit best here. We hope that R2 will concur that with the revisions the paper now has been improved. |
